# Atmospheric Pressure Plasma-Jet Treatment of PAN-Nonwovens—Carbonization of Nanofiber Electrodes

Andreas Hoffmann [1], Matthias Uhl [2], Maximilian Ceblin [2], Felix Rohrbach [3,4], Joachim Bansmann [5], Marcel Mallah [4], Holger Heuermann [3], Timo Jacob [2] and Alexander J. C. Kuehne [1,*]

[1]  Institute of Macromolecular and Organic Chemistry, Ulm University, Albert-Einstein-Allee 11, 89081 Ulm, Germany; andreas.hoffmann@uni-ulm.de
[2]  Institute of Electrochemistry, Ulm University, Albert-Einstein-Allee 47, 89081 Ulm, Germany; matthias.uhl@uni-ulm.de (M.U.); maximilian.ceblin@uni-ulm.de (M.C.); timo.jacob@uni-ulm.de (T.J.)
[3]  Institute for Microwave and Plasma Technology, FH Aachen—University of Applied Sciences, Eupener Str. 70, 52066 Aachen, Germany; rohrbach@fh-aachen.de (F.R.); heuermann@fh-aachen.de (H.H.)
[4]  Fricke & Mallah Microwave Technology GmbH, Werner-Nordmeyer-Straße 25, 31226 Peine, Germany; marcel.mallah@microwaveheating.net
[5]  Institute of Surface Chemistry and Catalysis, Ulm University, Albert-Einstein-Allee 47, 89081 Ulm, Germany; joachim.bansmann@uni-ulm.de
*  Correspondence: alexander.kuehne@uni-ulm.de

**Abstract:** Carbon nanofibers are produced from dielectric polymer precursors such as polyacrylonitrile (PAN). Carbonized nanofiber nonwovens show high surface area and good electrical conductivity, rendering these fiber materials interesting for application as electrodes in batteries, fuel cells, and supercapacitors. However, thermal processing is slow and costly, which is why new processing techniques have been explored for carbon fiber tows. Alternatives for the conversion of PAN-precursors into carbon fiber nonwovens are scarce. Here, we utilize an atmospheric pressure plasma jet to conduct carbonization of stabilized PAN nanofiber nonwovens. We explore the influence of various processing parameters on the conductivity and degree of carbonization of the converted nanofiber material. The precursor fibers are converted by plasma-jet treatment to carbon fiber nonwovens within seconds, by which they develop a rough surface making subsequent surface activation processes obsolete. The resulting carbon nanofiber nonwovens are applied as supercapacitor electrodes and examined by cyclic voltammetry and impedance spectroscopy. Nonwovens that are carbonized within 60 s show capacitances of up to 5 F g$^{-1}$.

**Keywords:** atmospheric plasma jets; carbonization; carbon nanofibers; electrodes

## 1. Introduction

Carbonization represents the second step during the conversion from a dielectric polymer precursor to carbon fibers [1,2]. Today, the majority of carbon fibers are produced from polyacrylonitrile (PAN)-precursor fibers that are first stabilized, usually between 250–320 °C in an oven in ambient atmosphere, before secondly being carbonized. Carbonization is typically performed at temperatures above 800 °C, regularly around 1300–1500 °C in the absence of oxygen, to avoid combustion. On a molecular level, the polymeric ribbons that have formed during stabilization fuse into graphitic sheets with nitrogen defects during carbonization. Depending on the precursor material and the choice of inert gas atmosphere during carbonization, the amount of nitride defects can be tuned [3–5]. These defects as well as the morphology of the material will influence the thermal and electrical conductivity of the carbonized fiber material [6–8]. The carbon material has graphitic as well as pyrolytic characteristics, where graphitic sheets in crystalline domains are either held together by strong van-der-Waals and π-π-interactions or by covalent linkages between graphitic sheets, respectively [9]. By contrast, carbonization of amorphous domains leads to turbostratic carbon, where graphite ribbons are crosslinked by individual,

more lose chains, leading to an overall disordered network structure [10]. Combined, this mixture of covalent and non-covalent interactions in amorphous and crystalline domains endows carbon fibers with their unprecedented mechanical properties of extremely high strength and high modulus at low density, as well as high thermal and electrical conductivity [11,12]. Because of these properties, carbon nonwovens represent ideal candidates as fiber-based electrodes.

Carbonization by thermal annealing can be an extremely slow process. For some types of precursor materials, carbonization can last for days, which entails high energy demand and cost for conventional thermal conversion processes [4]. Several alternative carbonization techniques have been developed to circumvent the drawbacks of thermal carbonization and to fine-tune new morphologies of carbon. Since stabilized carbon absorbs near-infrared (IR) light, thermally stabilized PAN-fibers can be carbonized using laser irradiation in the IR regime. This laser process transforms stabilized precursors into carbon-nanofiber nonwovens within seconds [13,14]. Alternatively, microwave irradiation at 2.45 GHz as well as microwave plasma treatment have been examined for carbonization of PAN-precursor fibers [15]. Both techniques significantly reduce the conversion time compared to conventional thermal processing [16,17].

Carbon fiber nonwovens represent interesting materials for electrodes in Li-batteries, fuel cells, and supercapacitors as well as for absorbers and filters [2,18–25]. The carbon nonwoven mats exhibit hierarchical porosity, with mesh sizes of the nonwoven in the range of micrometers, fiber diameters of the order of hundreds of nanometers, and a porous fiber surface on the sub and single digit nanometer scale [13,26]. This hierarchical porosity entails large surface areas and allows easy access for electrolyte and adsorbents to the smallest pores. Carbon fiber nonwovens are typically prepared by electrospinning or centrifugal spinning of a PAN-precursor solution onto a conductive collector, commonly aluminum foil [27]. After stabilization, the nonwoven is typically converted thermally to the carbon fiber nonwoven. Alternative carbonization techniques, such as microwave and plasma carbonization, require fully closed system to be able to control the atmosphere or produce low-pressure environments to ignite the plasma. Therefore, these processes cannot be scaled up easily to industrial throughputs and roll-to-roll processing. By contrast, non-equilibrium atmospheric pressure (AP) microwave plasma-jet technology represents a technique where the process gas (e.g., nitrogen, argon) at ambient pressures automatically generates an inert atmosphere around the stabilized non-woven to prevent its combustion. AP microwave plasma-jet technology is easily scalable, and jets can simply be parallelized by adding to existing system to extend the width, range, and throughput of the instrument [28,29]. However, to date, AP plasma-jet technology has not been employed for conversion to carbon materials.

Here, we employ non-equilibrium (also called non-thermal) AP nitrogen plasma-jet technology to carbonize thermally-stabilized PAN nanofiber nonwovens. We determine optimal carbonization conditions in view of their prospective applications as electrode materials with high electrical capacitance. We perform electrochemical characterization and find that the plasma-carbonized electrodes exceed conventionally carbonized nonwoven electrodes in their performance.

## 2. Materials and Methods

We conduct plasma jet carbonization on circular cutouts of thermally-stabilized PAN-nonwovens ($\emptyset$ = 2.5 cm, stabilized at $T_{max}$ = 280 °C), which are positioned in the intensity maximum in the center of the plasma jet with a circular cross-section. The plasma jet and the nonwoven are located in a chamber that is flooded by the inert process gas, to prevent oxidation or combustion in air (see Figure 1). We use a microwave plasma jet, which is powered by a 2.45 GHz magnetron (Fricke & Mallah Microwave Technology GmbH, Peine, Germany), with a maximum output power $P$ = 3 kW and nitrogen as process gas. AP plasma jets exhibit an axial intensity gradient [30,31]. Therefore, we vary the distance

*d* between nonwoven and plasma-jet nozzle to control the plasma jet intensity and the deposited energy exerted on the nonwoven.

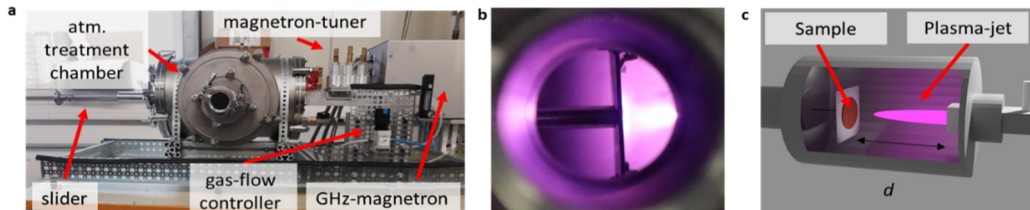

**Figure 1.** (**a**) Photograph of the atmospheric plasma-jet carbonization apparatus. Red arrows in (**a**) indicate the individual components. (**b**) Photograph inside the treatment-chamber, during plasma-jet carbonization. (**c**) Illustration of the plasma-jet carbonization setup, with the distance *d* between the sample and the plasma-jet nozzle.

## 3. Results and Discussion

### 3.1. Carbonization and Shrinkage of the Nonwoven

In the first step, we perform a series of measurements where we aim to find the optimal distance *d* between the sample and the plasma-jet nozzle. For these experiments of decreasing *d* (at $P$ = 2.4 kW), we observe bending of the nonwoven from flat discs to bowl-shaped geometries (Figure 2a). This deformation can be explained by faster carbonization and shrinkage on the frontside of the sample than on the backside. Usually, carbonization is accompanied by axial and radial fiber shrinkage, which in our case leads to a stronger shrinkage on the plasma facing frontside and bending of the nonwoven towards the nozzle of the jet. In fact, bending eventually leads to shorter distances of the edge region of the sample to the plasma jet than the center of the sample. The closer the distance to the plasma jet, the stronger the deformation (see Figure 2a). To evaluate this carbonization process and the effect of bending, we determine the conductivity of the samples in the center and at the edge as a proxy for the degree of carbonization, as shown in Figure 2b (see Figure S1 for current-voltage curves). While samples treated at distances $d \geq 45$ mm are not conductive, shorter distances *d* of 35 mm and 30 mm lead to slightly conductive edges. However, in these two samples, the center remains non-conductive. At a distance of 25 mm, we measure the highest conductivity at the edge of the bowl-shaped nonwoven (60 mS m$^{-1}$), with the center being weakly conductive as well (0.01 mS m$^{-1}$) (see Figure 2b). For even shorter distances between the nonwoven and the plasma jet nozzle, we observe occasional arc discharges between nozzle and the metallic sample holder, which leads to holes in the nonwoven. We therefore consider distances below 25 mm as not applicable.

To gain further insights into the plasma-jet driven carbonization, we conduct x-ray photoelectron spectroscopy (XPS) and Raman spectroscopy of the most conductive sample (carbonized at d = 25 mm). C1s-XPS spectra show that the edges of the nonwoven contained a greater amount of $sp^2$ carbon (81%), than the center of the nonwoven (74%) (see Figure 2c). Raman spectra show a high value for the intensity ratio of the D and G bands $I_D/I_G = 1.5$ for the center of plasma-jet carbonized nonwoven, while the edges give relatively low $I_D/I_G = 1.0$, reflecting the larger degree of carbonization (thermally carbonized samples contain a $I_D/I_G = 1.1$) and presence of more defect-free carbon at the edge of the treated samples (see Figure S2) [32]. These results corroborate our conductivity measurements, indicating incomplete carbonization in the center. The main cause for this inhomogeneity in carbonization is explained by the Gaussian power profile of the plasma jet, with greater power in the center than in the periphery. This power profile leads to faster carbonization, shrinkage, and therefore bending of the samples towards the plasma jet, as explained above.

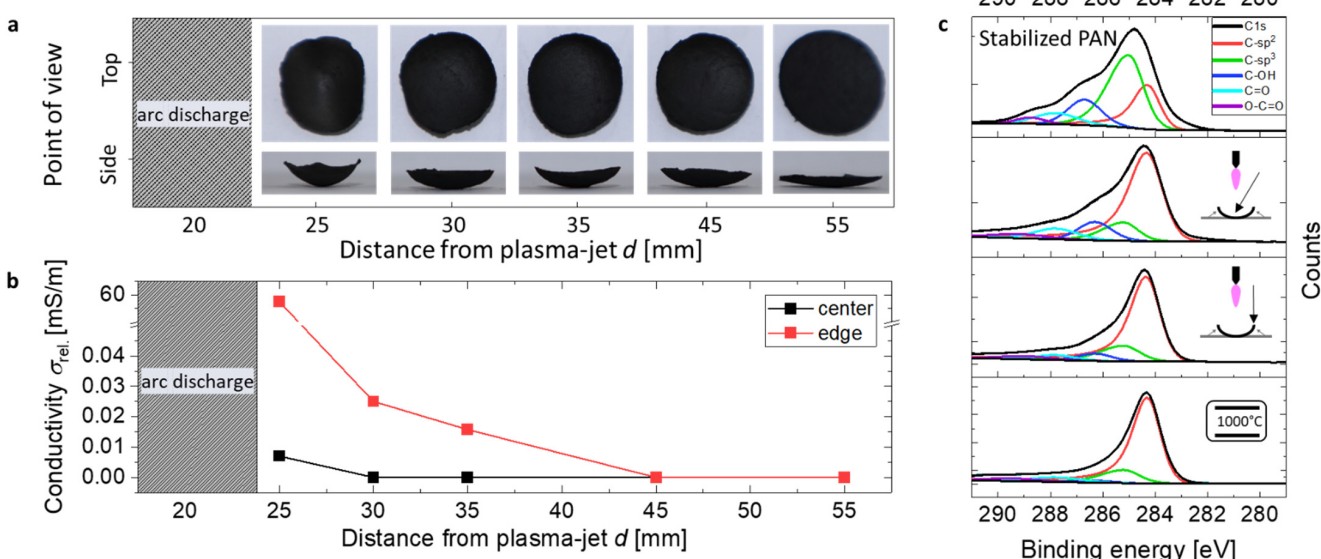

**Figure 2.** (**a**) Photographs of stabilized PAN-nonwoven, after plasma jet treatment at various plasma jet nozzle—nonwoven distances. (**b**) Conductivities of the samples from a), measured for the center (black) and the edge (red) of the nonwoven. (**c**) C1s-XPS spectra of PAN nonwoven after stabilization at 280 °C, the central part of plasma jet treated nonwoven at d = 25 mm, the edge of a plasma jet treated nonwoven at d = 25 mm and PAN-nonwoven after carbonization at 1000 °C by convective heat.

To overcome the problem of incomplete carbonization in the center and reduce bending of the sample, we perform consecutive front and backside plasma-jet exposure. This way we hope to induce back-bending, leading to more evenly carbonized samples with a flat geometry that are more useful for further processing and integration into electrochemical devices (see Figure 3).

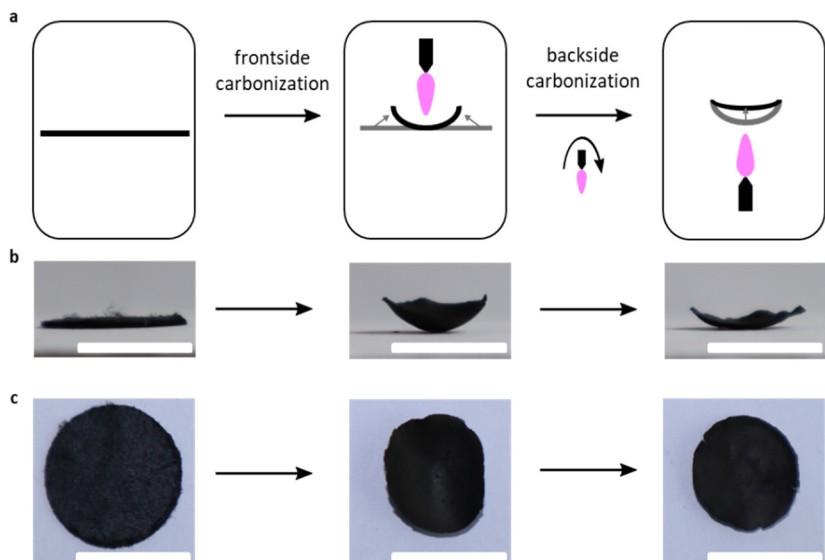

**Figure 3.** (**a**) Illustration of a two stepped plasma-jet carbonization, during which the nonwoven bend towards the plasma jet. (**b**) Photographs of the side-profile and (**c**) the top-profile of pristine, front carbonized and subsequently back-carbonized nonwoven. The scale bars indicate 10 mm.

Here, we flip the bowl-shaped nonwovens around and position the bowl-minimum in the intensity maximum of the plasma jet (see Figure 3a). We observe that the non-wovens bend back into an almost flat shape, after the subsequent second AP plasma-jet

carbonization of the nonwoven backside (see Figure 3b). We observe an overall shrinkage of 25%, which is a typical value also during thermal carbonization of PAN-nonwovens (see Figure 3c). Furthermore, the plasma-jet carbonized samples are brittle. Similarly, thermally carbonized nanofiber meshes exhibit the same lack of mechanical properties. XPS-measurements confirm that the nonwoven backside has been plasma-jet carbonized to the same degree as the center, raising and homogenizing the atomic ratios of carbon on the edge and in the center to 84% (see Table S2). The conductivity in the center and the edge are identical, while the conductivity of the front and backsides vary between 150 and 250 S/m depending on the plasma-jet exposure dose. While homogeneity in the extent of carbonization and conductivity can be restored after the second AP plasma-jet exposure, consecutive exposure might not be ideal for much larger samples of nonwovens. One workaround might be a simultaneous plasma-jet carbonization from both sides, which could prevent bending of the nonwoven during treatment.

### 3.2. Plasma-Jet Carbonized PAN-Nonwoven as Electrodes

A plasma jet consists of highly energetic electrons and ions. Therefore, plasma jets are often employed to ablate and roughen surfaces of the exposed material [33]. Such plasma-induced roughening has been reported for large fiber tows; however, roughening is usually deemed detrimental to the mechanical properties and the envisioned applications [16]. By contrast, surface roughening will be beneficial for electrochemical applications of carbon nanofiber nonwovens. Therefore, we investigate the surface of plasma-jet carbon nanofiber nonwovens by applying them as double-layer capacitor electrodes. We perform cyclic voltammetry and impedance spectroscopy on carbon nanofiber nonwovens, which have been carbonized from both sides by an AP plasma-jet with exposure times between $t = 5$–180 s and $d = 25$ mm (see Figure 4).

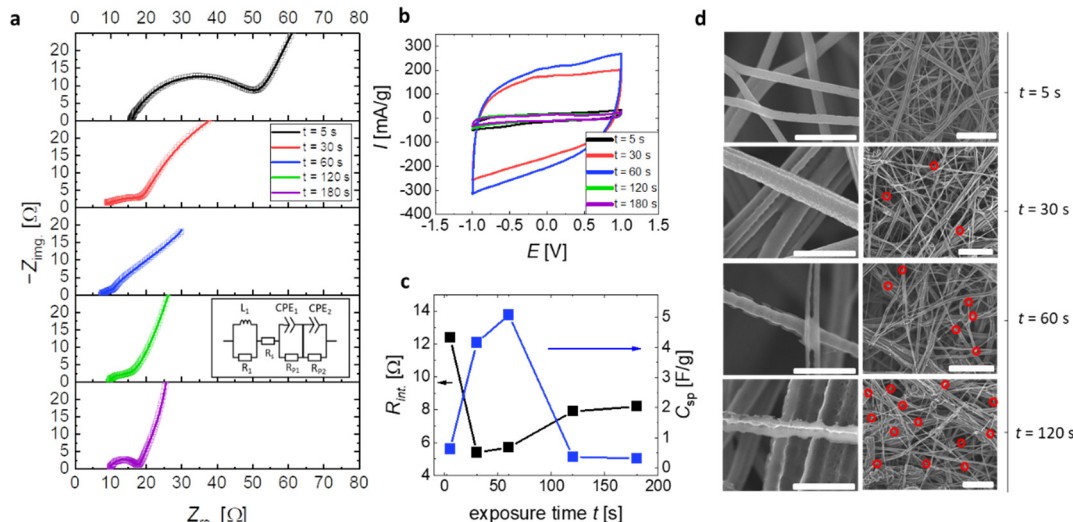

**Figure 4.** (**a**) Nyquist-plots with raw data points as hollow squares, fit as solid line and the used equivalent circuit in the inset, (**b**) cyclic voltammograms at 50 mV/s with the mass weighted current I and the electric potential vs. Ag-wire E and (**c**) specific capacitance $C_{sp}$ and internal resistance $R_{int.}$ of carbon nanofiber nonwoven, carbonized for various durations. (**d**) Close-up and wide-angle SEM-images of carbon-nanofiber nonwoven, carbonized for various plasma-jet exposure times. Red circles highlight breakage points of individual fibers. The scalebars indicate 2 μm (left row in **d**) and 20 μm (right row in **d**). Electrochemical measurements in (**a**–**c**) were conducted with the ionic liquid 1-Ethyl-3-methylimidazolium bis(trifluoromethylsulfonyl)imide, which was found to fully wet the nonwovens with a contact angle of 0°.

We fit the impedance data with the equivalent circuit, shown in the inset of Figure 4a (see Table S1 for the fitting parameters). The Nyquist plots contain a semicircle at high

frequencies followed by a non-vertical increase at decreasing frequencies, which can be interpreted as another diffusive process of ion transport into the pores of the electrode. Both processes show non-ideal RC behavior as a function of treatment time, and thus different diffusion limitations [34–38]. The Nyquist plots at different treatment times become steeper at low frequencies for longer plasma-jet exposure, indicating they become more capacitive. Furthermore, this increase may also indicate an improved ion diffusion (of both semicircle processes) and therefore a larger active surface of the nonwoven electrodes. The series of elements in the equivalent circuit is also corroborated by galvanostatic discharge curves, where we observe a slight *iR* drop in voltage upon discharging and an otherwise typical shape for diffusion controlled double layer capacitance (Figure S3). To test this hypothesis of ion diffusion in the porous carbon electrodes, we examine the fibers by scanning electron microscopy (SEM). We observe that longer carbonization times result in fiber surfaces with greater roughness (see Figure 4d). Fibers that have been exposed for only a short time have isolated pores on the fiber surface, while fibers that have been carbonized for longer periods of time have a rougher surface with a greater number of pores. This roughening of the surface can go as far as complete disintegration of the fiber surface and its typical round cross-section (compare Figure 4d $t = 5$ s versus $t = 120$ s). We also extract the resistivities of our electrodes from the fitting parameters of the spectra, shown in the Nyquist plots. (See Table S1 and Figure 4a). The resistance at the highest frequency corresponds to the internal resistance as also determined previously for similar EDCL electrodes [34,39]. Assuming that our system shows similar processes, we extract the internal resistance $R_{int}$ from our fitting parameter $R_s$ plotted for various plasma-jet exposure times $t$ (cf. Table S1 and Figure 4c). The electrode resistance seems to decrease for longer carbonization times, reaching a minimum at $t = 30$ s, and it increases again for durations $t > 60$ s. A critical loss of fiber percolation can explain this behavior by breaking of the fibers and, therefore, a loss of conductivity for exposure times $t > 30$ s (see Figure 4d). At $t = 60$ s, there seems to be an optimal balance with sufficiently high surface porosity and only a few fiber breaking points, which are not yet critical to the overall conductivity of the nonwoven electrode. Cyclic voltammograms show that the sample with $t = 60$ s has the highest specific capacitance of $C_{sp} = 5$ F g$^{-1}$ ($C_{sp} = 28$ mF cm$^{-2}$) at a scan rate of 5 mV/s (see Figures 4b and S4). We compare our nonwoven electrode, which has been carbonized by the non-equilibrium microwave AP plasma-jet, to a thermally carbonized nonwoven electrode (heating rate $\Delta T = 5$ K min$^{-1}$ to $T = 1050$ °C; $t_{hold} = 30$ min; $V_{N2} = 2$ L min$^{-1}$). The capacitance of the nonwoven carbonized by plasma-jet is higher by a factor of 25 compared to the thermally-carbonized carbon nonwoven electrode (cf. Figure S4).

## 4. Conclusions

We have developed a new AP microwave plasma-jet treatment for rapid carbonization to carbon fiber nonwovens for electrochemical charge storage applications. Double-sided treatment leads to flat and highly carbonized electrode architectures of tunable porosity. In combination with a plasma-jet stabilization process for PAN-precursor nonwovens developed by us previously, we can conduct the entire two-step conversion process using AP plasma-jets towards carbon nanofiber nonwovens. This process is less time- and energy-consuming than established heat convective processes of thermal stabilization and carbonization. The plasma-jet carbonization process will reduce the cost of processing carbon fiber nonwoven electrodes and absorbers. Furthermore, the morphology and porosity can easily be tuned towards the desired and required properties and porosities, making subsequent activation steps obsolete. In the future, studying the AP plasma-jet carbonization and activation processes separately will give further insights into the parameter and performance space that plasma-jet treatments provide.

**Supplementary Materials:** The following supporting information can be downloaded at: https://www.mdpi.com/article/10.3390/c8030033/s1, Figure S1: Current-voltage curves of plasma-carbonized nonwoven; Figure S2: Comparison in electrochemical performance of thermally carbonized nonwoven and plasma-jet carbonized nonwoven; Figure S3: Rama spectra of thermally car-

bonized nonwoven and plasma-jet carbonized nonwoven; Figure S4: Galvanostatic charge/discharge curves of plasma-jet carbonized nonwoven; Scheme S1: Cell-setup for electrochemical measurements; Table S1: Fitting parameters of the equivalent circuit fit in the of the Nyquist plots in Figure 4a. for various plasma-jet exposure times; Table S2: Atomic ratios of plasma jet carbonized samples, extracted from XPS-measurements; Table S3: Ratio of deconvoluted peak-areas in Figure 3c.

**Author Contributions:** A.J.C.K. devised and supervised the project. M.U., M.C. and T.J. performed and analyzed the electrochemical experiments. J.B. performed XPS. F.R., M.M. and H.H. designed and build the plasma-jet instrument. A.H. conducted all other experiments. A.H. and A.J.C.K. evaluated the data and wrote the first draft of the manuscript. All authors have read and agreed to the published version of the manuscript.

**Funding:** We thank the German Federal Ministry for Education and Research (BMBF) for financial support via the "KMU-innovativ" programs and the SuperCarbon project (project no. 13XP5036E). This work contributes to the research performed at CELEST (Center for Electrochemical Energy Storage Ulm-Karlsruhe) and we thank the German Research Foundation (DFG) for funding under Project ID 390874152 (POLiS Cluster of Excellence), and for funding within the priority program SPP 2248 Polymer-based Batteries (Project ID 441209207) and collaborative research center SFB 1316 Transient atmospheric pressure plasmas: from plasmas to liquids to solids (Project ID 327886311).

**Institutional Review Board Statement:** Not applicable.

**Informed Consent Statement:** Not applicable.

**Data Availability Statement:** Data is contained within the article and Supplementary Materials.

**Acknowledgments:** We thank Christian Herbert and Andreas Wego from Dralon GmbH for providing PAN. We thank Jerome Ogrzall from Fricke & Mallah Microwave Technology GmbH for constructing the plasma carbonization chamber. We thank Bastian Beitzinger for Raman measurements.

**Conflicts of Interest:** The authors declare no conflict of interest.

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
