# Peer review of "Atmospheric Pressure Plasma-Jet Treatment of PAN-Nonwovens—Carbonization of Nanofiber Electrodes"

_carbon, 2022_

Round 1

Reviewer 1 Report

This paper reports the use of atmospheric pressure plasma-jet treatment for the rapid carbonization of PAN nanofibers. The authors examined the carbonized nonwovens as supercapacitor electrode and their properties by CV and impedance spectroscopy. Even though the method allows very fast carbonization in just 60s with capacitance up to 5 F g-1. There are problems with the method. It does not produce the homogeneous carbonization of samples even for these porous materials. For the nonporous samples, I would expect completely different carbonization for the front side and backside. The mechanical properties of the materials should be tested and compared with thermally carbonized fiber samples. Will they be more brittle or not? Overall, the concept is useful in producing carbonized samples in short times. So I would recommend the paper after the following points are properly addressed during the revision.

·         Because of differences in peak intensity (284.8 eV – C-C), It is difficult to see the changes in the composition in C1s spectra. The peak intensity should be comparable to follow the changes in deconvoluted peaks. A table with deconvoluted data should accompany this in ESI as well.

·         There are two Fig. 1.. The latter should be Fig. 2.

·         Page 4, Lines 120-135 discuss that it is impossible to produce completely homogenous carbonization. So this is the main drawback compared to the thermally carbonized samples. Even though the concept allows production carbonization within a short time, the resultant mats will suffer heterogeneous conductivity. These points should be discussed in the paper.

·         What happens if the PAN film is used? Will conductivity difference be higher or lower through the surface? Did the author take samples from the backside as well? I think it will have a completely different composition and conductivity than the front side. The authors need to check that! To me, these are the main cons of this technique.

·         Fig. 2 should be Fig. 3.

·         Does the collector stay constant or rotate?

·         ESI, page 1 “in DMSO/Acetone” what is the composition? Mw of PAN should be written too!

·         Additional information should be given for the electrospinning part.. What were the humidity, tip-to-collector distance, the speed of the drum collector, flow rate, and the applied voltage?

·         Regarding XPS measurements, what were the numbers of scans for survey and C1s?

·         Table S2, the atomic compositions of some samples do not reach 100%. Please check all data.

·         In the Materials section, “1-Ethyl-3-methylimidazolium bis(trifluoromethylsulfonyl)imide (EMIM TFSI; 99.5%) was obtained from Iolitec GmbH.” Where was this chemical used? I could not find it in the main text and ESI

·         The authors should perform mechanical analysis of the carbonized sample? At least they can show the optical photos during the folding and twisting of the carbonized fiber mats. 

Author Response

please see answers to reviewer 1 document 

Reviewer 2 Report

In this paper, the autors utilize an atmospheric pressure plasma-jet to conduct carbonization of stabilized PAN nanofiber nonwovens. The authorsd explore the influence of various processing parameters on the conductivity and degree of carbonization of the converted nanofiber material. Some of the comments need to be addressed before publication :

1. THe ID/IG ratio should be calculated and analysed for the all comparative samples from the Fig. S3.

2. What are the main causes behind the variation of Tthe RAMAN peak ration on thesame sampleat different place? Needs proper explanations.

3. The electrochemical data presented in the Fig. 4 (a, b, and c) are insufficient. The GCD of the as prepared samples also need to check for the proper utilization of the as prepared carbon materials for the electrochemical applications!!

4. In Fig. 4 (D) the morphology of the fiber is detorated at 60 s and 120 s, why? It may effect the physical properties  of the electrode materials.

5.The authors can use the following review paper for the citation in the introduction section: https://doi.org/10.3390/electrochem2020017 and  https://doi.org/10.1016/j.est.2021.103927

6. Author can add the stress strain properties , hydorphilic nature of electrode materials forthe firther elaboration of the materials.

Author Response

please see answers to reviewer 2 document for replies and indicated changes.

Reviewer 3 Report

A novel, comprehensive work on the carbonization of carbon nanofiber nonwovens by atmospheric plasma treatment. Work well conducted, conclusions supported by the data, and paper well written. I recommend the paper for publication as is.

Author Response

we thank the reviewer for taking the time to evaluate our manuscript.

Round 2

Reviewer 1 Report

The authors revised the manuscript according to my comments/suggestions. The manuscript can be accepted as it is.

Reviewer 2 Report

All the corrections are appended.